# Extended Cleavage Specificities of Two Mast Cell Chymase-Related Proteases and One Granzyme B-Like Protease from the Platypus, a Monotreme

**DOI:** 10.3390/ijms21010319

**Published:** 2020-01-02

**Authors:** Zhirong Fu, Srinivas Akula, Michael Thorpe, Lars Hellman

**Affiliations:** Department of Cell and Molecular Biology, Uppsala University, Uppsala, The Biomedical Center, Box 596, SE-751 24 Uppsala, Sweden; fuzhirong.zju@gmail.com (Z.F.); Srinivas.Akula@icm.uu.se (S.A.); getmeinahalfpipe@gmail.com (M.T.)

**Keywords:** platypus, monotremes, mast cell, chymase, human chymase, cleavage specificity, animal model

## Abstract

Mast cells (MCs) are inflammatory cells primarily found in tissues in close contact with the external environment, such as the skin and the intestinal mucosa. They store large amounts of active components in cytoplasmic granules, ready for rapid release. The major protein content of these granules is proteases, which can account for up to 35 % of the total cellular protein. Depending on their primary cleavage specificity, they can generally be subdivided into chymases and tryptases. Here we present the extended cleavage specificities of two such proteases from the platypus. Both of them show an extended chymotrypsin-like specificity almost identical to other mammalian MC chymases. This suggests that MC chymotryptic enzymes have been conserved, both in structure and extended cleavage specificity, for more than 200 million years, indicating major functions in MC-dependent physiological processes. We have also studied a third closely related protease, originating from the same chymase locus whose cleavage specificity is closely related to the apoptosis-inducing protease from cytotoxic T cells, granzyme B. The presence of both a chymase and granzyme B in all studied mammals indicates that these two proteases bordering the locus are the founding members of this locus.

## 1. Introduction

Mast cells (MC) are hematopoietic cells distributed along both external and internal surfaces of the body where they most likely act as a first line of defence [1,2,3]. They are tissue resident cells that are frequently found in connective tissue of the skin and around blood vessels and nerves as well as in the mucosa of the airways and intestines. MCs pre-store a number of inflammatory mediators in cytoplasmic granules. These mediators are rapidly exocytosed from the cell following activation triggered by various stimulators, including cross-linking of receptor-bound IgE, anaphylatoxins (C3a, C4a and C5a) and substance P. The mediators released from MCs include histamine, heparin, various proteases, prostaglandins and leukotrienes. Histamine, heparin and proteases are granule-stored whereas leukotrienes and prostaglandins are produced from arachidonic acid upon cell stimulation and are not granule-stored. The majority of proteins found in the MC granules are serine proteases [4,5,6]. These proteases can generally be sub-divided into chymases and tryptases [7,8]. Chymases are chymotrypsin-like and cleave substrates after aromatic amino acids. Phylogenetic analyses of the chymases have led to the identification of two distinct subfamilies, the α-chymases and the β-chymases (Figure 1) [9,10,11,12]. The α-chymases are found as a single gene in all species investigated, except for ruminants, where two very similar α-chymase genes have been identified [12,13]. β-chymases have only been identified in rodents, and as single β-chymase-like genes in dogs and cats [12]. Interestingly, the rodent α-chymases mouse MC protease (mMCP)-5, rat MC protease (rMCP)-5 and hamster chymase II have changed their primary cleavage specificities from aromatic amino acids (chymotrysin-like) to aliphatic amino acids (elastase-like) [14,15,16,17].

In mammals, the mast cell chymotryptic enzymes are found in one chromosomal locus, the chymase locus. In humans, this locus encodes four genes: one MC expressed enzyme, the α-chymase; one neutrophil and MC expressed protease, cathepsin G; and two T-cell granzymes, granzymes (gzm) H and B (Figure 2) [12]. This locus is present in all studied mammals and related enzymes have also been identified in the American alligator and in the clawed frog, *Xenopus laevis* [12]. However, no closely related members of this locus have been found in fishes or birds [12]. The chymase locus also has the same bordering genes in all mammals studied from opossums to humans; at one end by the mast cell α-chymase and at the other end by granzyme B (Figure 2). However, there have been massive changes in gene numbers in some placental mammals within these borders, primarily in rodents but also in ruminants. As previously described, the human locus contains four active serine protease genes: the chymase, cathepsin G and two granzymes, B and H. Both mice and rats have experienced large increases in gene numbers in this locus, most likely by successive gene duplications. Mice have 15 active serine protease genes and rats have 28 such genes [12]. All of the studied mammals, from marsupials to placental mammals, have a classical chymotryptic enzyme expressed by mast cells, except for the rabbit and the guinea pig, where the chymases have become restricted in their substrate selectivity to become strict Leu-ases [18,19]. All of these chymotryptic enzymes, and the Leu-ases are encoded from the chymase locus of the respective species. 

To look deeper into the conservation of the chymase locus and the presence and conservation of classical MC chymases, here our interest lies with the monotremes, which are an early branch of egg-laying mammals. This branch of the mammalian tree has been estimated to have separated from the common mammalian branch around 220 million years ago. Today the monotremes are only represented by three extant members, the platypus as well as the long- and short-nosed echidnas. The platypus genome is still incomplete and currently we have found three genes, which cluster closely in a phylogenetic tree with the placental and marsupial chymase loci genes (Figure 1). The three genes are found in two separate contigs where two of these, named platypus granzyme B (GzmB) and platypus DDN1-like, cluster together with the chymases, and one gene called platypus granzyme BGH (GzmBGH), clusters with the granzymes. When looking at the three major specificity-determining amino acids, which sit in the substrate binding pocket (based on crystallized 3D structures), we find that the two first platypus enzymes have very similar triplets as the previously characterized classical chymases (SGN and SVN, compared with SGA). The third, platypus granzyme BGH, has a triplet that is similar to the granzyme Bs of several placental mammalian species, AGR, indicating that it has asp-ase activity similar to granzyme B of placental mammals. To obtain a better view of the origin and evolution of MC chymases and granzyme B during vertebrate evolution, we have studied the extended cleavage specificity of these three platypus enzymes. We can show that both platypus ‘granzyme B’ (the chymase) and platypus DDN1-like are classical chymases with specificities very similar to human and mouse MC chymases, whereas the platypus GzmBGH is an asp-ase with very similar specificity to granzyme B of placental mammals. This shows that these enzymes have been conserved for more than 200 million years of mammalian evolution, which gives strong indications for their importance in the immune functions of MCs and cytotoxic T cells. 

## 2. Results

### 2.1. Production, Activation and Purification of Recombinant Platypus Enzymes 

The coding regions for the three separate chymase locus-related platypus enzymes were ordered as designer genes from Genscript and inserted in the mammalian expression vector pCEP-Pu2 [20]. These three clones were then transfected into the human embryonic kidney cell (HEK) 293-EBNA for expression. After establishing confluent cultures of the transfected cells, the medium was collected and the secreted protein was purified on Ni-chelating IMAC columns. After analysis of the protein fractions from the purification, all three clones gave sufficient amounts of protein for further analyses (Figure 3). After purification, the proteins were activated by removing the N-terminal His-6 tag and the enterokinase site by cleavage with enterokinase. The active protein (i.e., enterokinase cleaved) is approximately 1.5 kDa smaller than the uncleaved protein (no enterokinase addition) (Figure 3).

### 2.2. Chromogenic Substrate Assay and Phage Display

A panel of 11 different chromogenic substrates was used to try to determine the primary specificities of two of these enzymes, GzmB and Gzm BGH. However, no activity was detected with any of the 11 substrates with these two enzymes indicating high specificities. We also tried several times with substrate phage display without success with all three of these enzymes. A similar phenomenon has been observed for a few other hematopoietic serine proteases, which later have been found to be highly specific. This further supports the indication from the chromogenic substrate assay that all enzymes are more specific than the other mammalian enzymes previously investigated.

### 2.3. Analysis of the Extended Cleavage Specificity by the Use of Recombinant Protein Substrates

As neither chromogenic substrates nor phage display resulted in any information concerning the extended specificities of these three platypus enzymes, we instead tried a third alternative; to use a new type of recombinant substrate. A panel of such substrates has previously been designed and produced for the verification of the extended cleavage specificity following phage display analysis for a number of previously analysed mammalian hematopoietic and coagulation serine proteases [17,21,22,23,24,25,26,27,28,29,30]. In this type of substrate, the consensus sequence obtained from the phage display analysis is generally used, often a region of approximately nine amino acids. This region is inserted into a linker region between two *E. coli* thioredoxin (Trx) molecules by ligating a double stranded oligonucleotide encoding the actual sequence into a *BamHI* and a *SalI* site of the vector construct (Figure 4A). For purification purposes we also inserted a His-6 tag to the C-terminal of the second Trx molecule of this protein (Figure 4A). A number of related and unrelated substrate sequences were also produced with this system, by ligating the corresponding oligonucleotides into the *BamHI/SalI* sites of the vector. All of these substrates were expressed as soluble proteins in a bacterial host, *E. coli* rosetta gami, and purified on IMAC columns to obtain a protein with a purity of 90–95%. We have produced more than 270 such substrates for the analysis of other serine proteases and selected a few such with different types of amino acids, including aromatic, aliphatic, negatively or positively charged amino acids in the central position of the substrate for an initial screening of the three platypus serine proteases. 

We started the analysis with the two enzymes expected to have chymotryptic activity: the platypus GzmB and platypus DDN-1-like. For these two enzymes, we selected a panel of different chymase substrates as well as an elastase and a tryptase substrate as negative controls. Both of these two enzymes showed a relatively strict chymotryptic activity. The best activity for both of these very closely related enzymes was seen with the substrate VVLF↓SGVL (arrow indicating the potential cleavage site) (Figure 4 and Figure 5). A slight drop in cleavage activity was seen when a negative charge was introduced in the P1´or P2´positions. In contrast, by introducing a positive charge in the P2 position, we observed an almost complete block in cleavage (Figure 4 and Figure 5). Furthermore, almost no cleavage was seen for substrates with a Trp or a Leu in the P1 position (Figure 4 and Figure 5). These two enzymes did not display any tryptase activity as substrates with an Arg in the P1 position were not cleaved (Figure 4 and Figure 5). Changing the P1’position from a Ser to a Leu also resulted in a dramatic reduction in cleavage, indicating the importance of the amino acid in this position. An Arg in the P1´position was more favoured but still considerably less than Ser in this position (Figure 4 and Figure 5). These two enzymes showed very similar cleavage patterns, which was expected due to their very high similarity in primary sequence. However, a few minor differences could be observed. One such difference was that platypus DDN-1-like tolerated a negatively charged residue in the P1’position better than granzyme B (Figure 4 and Figure 5).

The names of the platypus proteases do not reflect other close mammalian homologs and have been generated by early sequence similarity analyses of few members of this subfamily of proteases. The platypus GzmB is not the classical granzyme B homolog but instead the platypus MC chymase. Currently, this is somewhat confusing but it may take some time before the name will change based on the available evidence.

As a next step, we analysed the extended cleavage specificity of the platypus GzmBGH, the enzyme with a granzyme B-like substrate pocket. The substrates with a negatively charged amino acid in the P1 position were the only ones that showed any cleavage activity (Figure 6). This enzyme was found to prefer sequences with a Phe preceding the P1 Asp and disliked positively charged amino acids like Arg in this position, the P2 position (Figure 6). An Arg in the P2’ was also negatively influencing cleavage (Figure 6). The most optimal sequence in this analysis was LIE^↓^FD^↓^VFVQ (arrows indicate potential cleavage sites), a sequence with two negatively charged residues, one in the P1 and one in the P3 position and an aromatic amino acid in the P2 position (Figure 6).

## 3. Discussion

The chymase loci of all previously analysed mammals, including both placental mammals and marsupials, are bordered at one end by an α-chymase and at the other end by granzyme B [12]. Based on the analyses of two genomic contigs in the platypus, which encode enzymes that cluster closely with other mammalian chymase loci genes, we can now conclude that these platypus enzymes also most likely represent these bordering genes. On one of these contigs, there are two very closely related chymases, which both appear to be α-chymases, indicating a relatively recent gene duplication, and on the second contig there is an asp-ase closely related to the other mammalian granzyme Bs. Therefore, all of the major extant mammalian lineages have the same general pattern with a classical MC chymase and a granzyme B [12]. The platypus is so far the only species analysed where the locus has been separated into two regions which are located on two different chromosomes (Figure 2). In the platypus to chromosomes 13 and 14, where only the chymases on chromosome 13 have previously identified bordering genes related to the chymase locus of other mammals (Figure 2). Interestingly, in the majority of rodents the α-chymases have changed primary specificities and become elastases. However, their lack of a chymotryptic α-chymase seems to have been compensated for by the presence of one or several chymotryptic β-chymases. All the rodents that have elastolytic α-chymases, including mice, rats and golden hamsters, have mainly chymotryptic β-chymases [14,15,16,17]. In this locus, all the analysed mammals also have granzyme B or a granzyme B-like protease with asp-ase activity, which indicates a strong conservation of both a MC chymotryptic enzyme and of an apoptosis-inducing granzyme B in cytotoxic T cells [12,28]. Currently, the only two species which differ from this pattern are the rabbit and guinea pig, where the β-chymase or α-chymase, respectively, have become strict leu-ases. Furthermore, they both appear to lack a compensating chymotryptic enzyme encoded from the chymase locus [19]. The follow-up question is what central targets these enzymes have and how the rabbit and guinea pig can compensate for this loss of a chymase in MCs. One possibility is that another locus is compensating for this loss possibly by gene duplication and a change in tissue specificity. A similar situation has occurred in pigs, sheep and cows where genes within the chymase locus have duplicated, most likely a granzyme or a cathepsin G gene, and that these new copies have changed tissue specificities, from expression in hematopoietic cells to primarily expression in the duodenum of the intestinal region; therefore, they are named duodenases [12,31,32]. However, no analyses of the protease content of rabbit or guinea pig mast cells have yet been performed to screen for chymotryptic enzymes originating from other protease loci, which could substantiate or disprove such a possibility. 

A number of potential targets for the MC chymotryptic enzymes have been identified, including angiotensin I cleavage in the context of blood pressure regulation, cleavage of snake or scorpion toxins in the protection against their toxic effects, cleavage of fibronectin and the activation of pro-collagenases in connective tissue turnover, cleavage of thrombin to regulate coagulation and several additional potential targets (reviewed in [11,33]). One question is how the rabbit and guinea pig have solved these functions if they lack a MC chymotryptic enzyme? One interesting finding, which touches on one of these problems, is the major chymotryptic enzyme of rat connective tissue MCs, rMCP-1. This enzyme has been shown to primarily activate angiotensin I, by cleavage at Phe-8 to form angiotensin II but also has the ability to cleave at Tyr-4, which leads to inactivation of angiotensin I. The rat has an extra β-chymase gene that is a potent angiotensin I converter, which has changed tissue specificity and is now primarily expressed in vascular smooth muscular cells, where it can activate angiotensin independently of mast cells. This enzyme has therefore been named vascular chymase [34,35]. In our minds, this supports the role of MC chymotryptic enzymes in blood pressure regulation. When this function has been lost from MCs, it has to be compensated for by another tissue, and in this case also another enzyme. Another serine protease locus may have duplicated copies of a chymotryptic enzyme that is either expressed in rabbit and/or guinea pig MCs, or that such enzymes are expressed in other cell types in tissues where MCs normally reside.

Another striking observation in this study was the lack of cleavage of the chromogenic substrates and also the difficulty in obtaining good amplification during phage display pannings of the platypus enzymes. We have observed this phenomenon with several related enzymes from non-mammalian species including enzymes from both *Xenopus laevis*, an amphibian, and the Chinese alligator, a reptile. Two of these enzymes have a specificity determining triplet in their predicted substrate binding pocket of the enzyme, which strongly indicates asp-ase specificity, similar to the granzyme Bs of mammals, and the third, an alligator enzyme, has a triplet that is identical to primate cathepsin G (Figure 1). However, when using a panel of chromogenic substrates and phage display, we also saw no cleavage and no increase in titers of phages after the pannings, similar to what we observed with the platypus enzymes (unpublished data). However, in contrast to the platypus where the recombinant substrates were cleaved, we have not had any success with the reptile and amphibian supposedly chymase locus genes. This is true of all the recombinant substrates currently analysed, which includes a large selection of over 270 different possible substrates. We have also analysed a few hematopoietic fish proteases and generally it seems that the mammalian enzymes have broader specificities and thereby can be more easily studied with chromogenic substrates and phage display in comparison to their reptile, amphibian and fish homologues [27]. Currently, we have no good explanation to this phenomenon except perhaps the extended specificities are so specific that they are not picked up against classical granzyme B substrates. Even minor differences in substrate sequences compared to the proteases’ consensus may result in a complete loss of cleavage. Similarly, the phage library used may also have very low numbers of clones with such specifically preferred sequences and therefore the phage pannings do not result in amplification of any/enough clones compared to the background.

In summary, this study shows that the platypus has both a MC chymotryptic enzyme and a granzyme B homolog. This reveals that all three mammalian lineages, the (egg-laying) monotremes, the marsupials and the placental mammals, have these two enzymes in their immune repertoire, which is a very strong indication that they play essential functions in the immune system of mammals. Interestingly, there is no clear candidate for a chymase in the few chymase locus-related enzymes identified in reptiles and amphibians. However, we have identified enzymes that look like granzymes Bs in their potential substrate pockets, indicating that the apoptosis-inducing function was the first function of an enzyme within this locus and the chymase followed later. An alternative scenario is that the chymases have been lost in the two lineages and been replaced by an enzyme from another locus. This is possibly valid for the leu-ase expressing rabbit and the guinea pig; questions that we hope can be addressed in the near future.

## 4. Materials and Methods 

### 4.1. Production and Purification of Recombinant Platypus Enzymes

The sequences of the three platypus enzymes were retrieved from the NCBI database. cDNA copies of the mRNA were designed, based on the gene sequences, and ordered as designer genes from GenScript (Piscataway, NJ, USA). The individual designer gene was inserted into the mammalian expression vector pCEP-Pu2 and sequence verified. The construct contains a signal sequence, an N-terminal His-6 tag and an enterokinase site for purification and later activation by enterokinase cleavage. The vector was transfected into the human embryonic kidney (HEK) 293-EBNA cell line for expression of the recombinant enzyme. After purification on IMAC Ni ^2+^ agarose, the enzyme was activated by the addition of 1 µL enterokinase into 90 µL column eluate containing the recombinant protein. The sample was mixed, followed by a 5 h incubation at 37 °C to activate the protease. The purity and activation of the enzyme was determined by separation on 4–12% pre-cast SDS-PAGE gels (Invitrogen, Carlsbad, CA, USA). An amount of 2.5 µL sample buffer, containing sodium dodecyl sulfate (SDS), and 0.5 µL β-mercapto-ethanol was added to the 10 µL of protein sample followed by 85 °C heating, before separation on the SDS-PAGE gel. The gels were stained overnight in colloidal Coomassie staining solution followed by de-staining [36]. 

### 4.2. Generation of Recombinant Substrates for the Analysis of the Cleavage Specificity

A new type of substrate was originally developed to verify the results obtained from the phage display analysis. Two copies of the *E. coli* thioredoxin (Trx) gene were inserted in tandem into the pET21 vector for bacterial expression (Figure 4A). In the C-terminal end a His-6 tag was inserted for purification on Ni^2+^ IMAC columns. In a linker region, between the two thioredoxin molecules, the different substrate sequences were inserted by ligating double-stranded oligonucleotides into two unique restriction sites, one *BamHI* and one *SalI* site (Figure 4A). This linker region contains a kinker region of repeating Ser-Gly residues which forms a flexible region open for access by a protease to cleave. The sequences of the individual clones were verified after cloning by sequencing of both DNA strains. The plasmids were then transformed into the *E. coli* Rosetta gami strain for protein expression (Novagen, Merck, Darmstadt, Germany). A 10 mL overnight culture of the bacteria harbouring the plasmid was diluted 10 times in LB + Amp and grown at 37 °C for 1–2 h until the OD (600 nm) reached 0.5. IPTG was then added to a final concentration of 1 mM. The culture was grown at 37 °C for an additional 3 hours under vigorous shaking, after which the bacteria were pelleted by centrifugation at 3500 rpm for 12 minutes. The pellet was washed once with 25 ml PBS + 0.05 % Tween 20. The pellet was then dissolved in 2 mL PBS and sonicated 6 × 30 s to open the cells. The lysate was centrifuged at 13,000 rpm for 10 min and the supernatant was transferred to a new tube. Five hundred microliters of Ni-NTA slurry (50:50) (Qiagen, Hilden, Germany) was added and the sample was slowly rotated for 45 min at room temperature. The sample was subsequently transferred to a 2 mL column and the supernatant was allowed to slowly pass through the filter leaving the Ni-NTA beads with the bound protein in the column. The column was then washed four times with 1 ml washing buffer (PBS + 0.05 % Tween + 10 mM Imidazole + 1 M NaCl). Elution of the protein was performed by adding 150 µL elution buffer followed by five 300 µL fractions of elution buffer (PBS + 0.05 % Tween 20 + 100 mM imidazole). Each fraction was collected individually. Ten microliters from each of the eluted fractions was then mixed with 1 volume of 2× sample buffer and 1 µL β-mercapto-ethanol and then heated for 3 min at 80 °C. The samples were analysed on an SDS bis tris 4–12% PAGE gel and the second and third fractions that contained the most protein were pooled. The protein concentration of the combined fractions was determined by Bio-Rad DC Protein assay (Bio-Rad Laboratories Hercules, CA USA). Approximately 60 µg of recombinant protein was added to each 120 µL cleavage reaction (in PBS). Twenty microliters from this tube was removed before adding the enzyme, the 0 minute time point. The active enzyme was then added and the reaction was kept at room temperature during the entire experiment. Twenty microliter samples were removed at the indicated time points (15, 45 and 150 min) and stopped by addition of one volume of 2× sample buffer. One microliter β-mercapto-ethanol was added to each sample followed by heating for 3 minutes at 80 °C. Twenty microliters from each of these samples was then analysed on 4–12% pre-cast SDS-PAGE gels (Invitrogen, Carlsbad, CA, USA). The gels were stained overnight in colloidal Coomassie staining solution and de-stained for several hours according to previously described procedures [36].

## Figures and Tables

**Figure 1 ijms-21-00319-f001:**
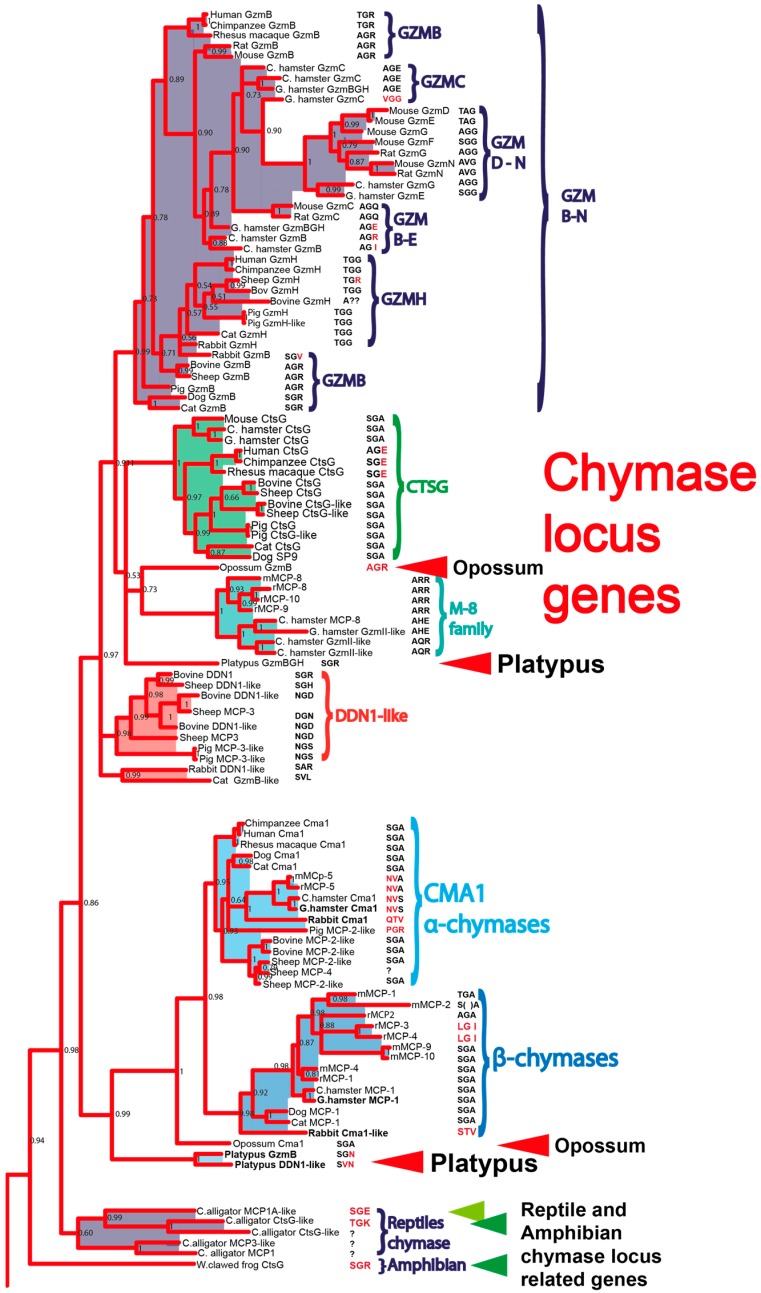
A phylogenetic tree of chymase loci encoded serine proteases. The amino acid sequences of a panel of chymase loci encoded proteases were analysed for sequence relatedness with the program MrBase. A bootstrap tree based on 1000 replicates was generated and the bootstrap values are depicted at each branch of the tree. The different subfamilies of chymase loci genes were colour-coded for easy identification and the genes of primary interest, monotreme and marsupial enzymes, are marked by red arrows and the related alligator and Xenopus proteases are marked with green arrows. The granzyme B related in dark green and the cathepsin G related in light green.

**Figure 2 ijms-21-00319-f002:**
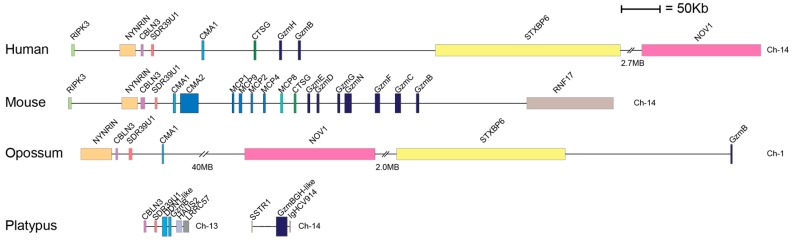
The chymase locus. The chymase locus encodes a number of hematopoietic serine proteases, including the α-chymases, β-chymases, cathepsin G, and several granzymes [12]. Genes are colour-coded: the α-chymase-related genes are marked in light blue, the β-chymases in slightly darker blue, cathepsin G in green, the M8 family in light green, the granzymes in dark blue.

**Figure 3 ijms-21-00319-f003:**
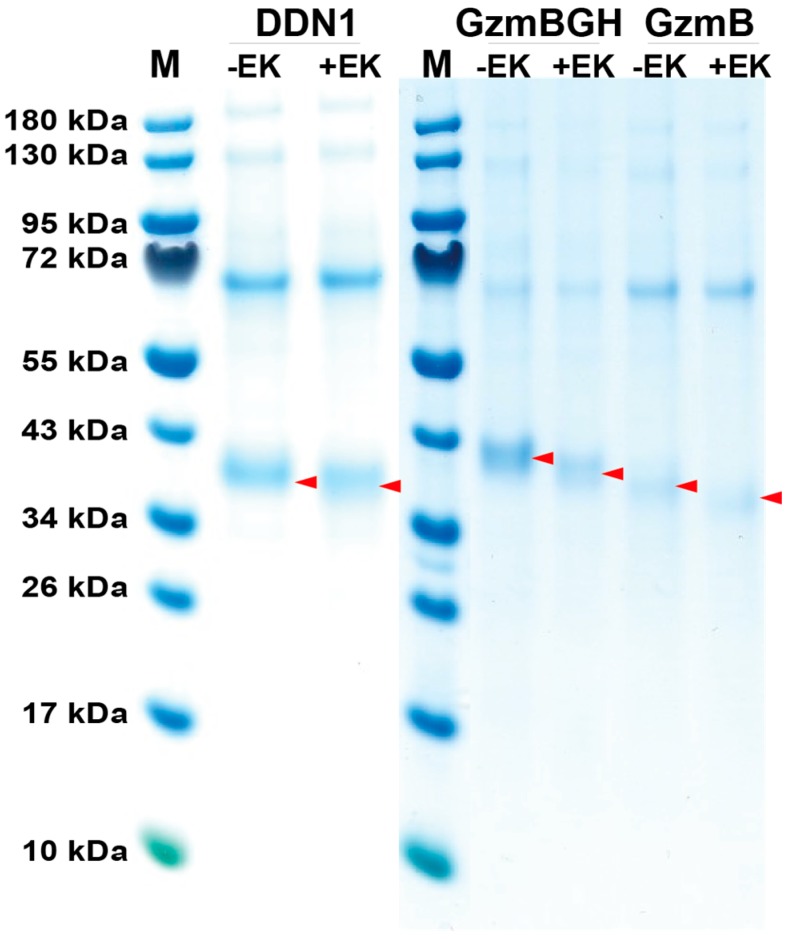
Analysis of the recombinant platypus enzymes used in the determination of their extended cleavage specificities. Three different recombinant platypus enzymes were produced as proenzymes, containing an N-terminal His_6_-tag and an enterokinase site, in the HEK-293 EBNA cells with the episomal vector pCEP-Pu2. These proenzymes were first purified on Ni-NTA beads (-EK) and then activated by removal of the His-6 tag by enterokinase digestion (+EK). After purification and activation, the three enzymes were analysed by separation on SDS-PAGE and visualized with Coomassie Brilliant Blue staining. The proteases before and after enterokinase cleavage are marked with small red arrow heads. The few additional bands seen on the gel are originating from the conditioned cell medium, and the major band of a size of approximately 69 kDa is bovine serum albumin from the serum that represent the major protein of the cell medium.

**Figure 4 ijms-21-00319-f004:**
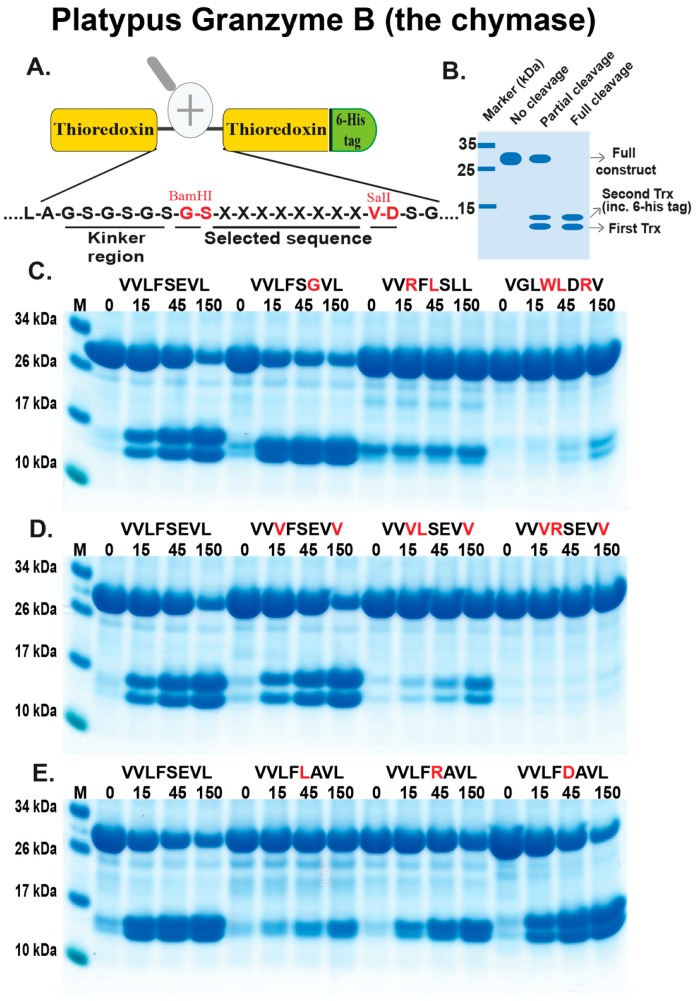
Analysis of the cleavage specificity of platypus GzmB (the chymase) by the use of recombinant protein substrates. Panel (**A**) shows the overall structure of the recombinant protein substrates used for analysis of the efficiency in cleavage by the three different enzymes (analysed in Figure 4, Figure 5 and Figure 6). In these substrates, two thioredoxin molecules are positioned in tandem and the proteins have a His-6 tag positioned in their C-termini of the second thioredoxin. The different cleavable sequences are inserted in a linker region between the two thioredoxin molecules by the use of two unique restriction sites, one *Bam HI* and one *SalI* site, which are indicated in the bottom of panel (**A**). In panel (**B**), an example cleavage is shown to highlight possible cleavage patterns. Panels (**C**–**E**) show the cleavage of a number of substrates by the platypus granzyme B, the chymase. The sequences of the different substrates are indicated above the images of the gels. The red residues highlight the major difference between this substrate and the reference substrate positioned at the beginning (Left side) of this panel. The time of cleavage, in minutes, is also indicated above the corresponding lanes of the different gels. The un-cleaved substrates have a molecular weight of approximately 25 kDa and the cleaved substrates appear as two closely-located bands with a size of 12–13 kDa.

**Figure 5 ijms-21-00319-f005:**
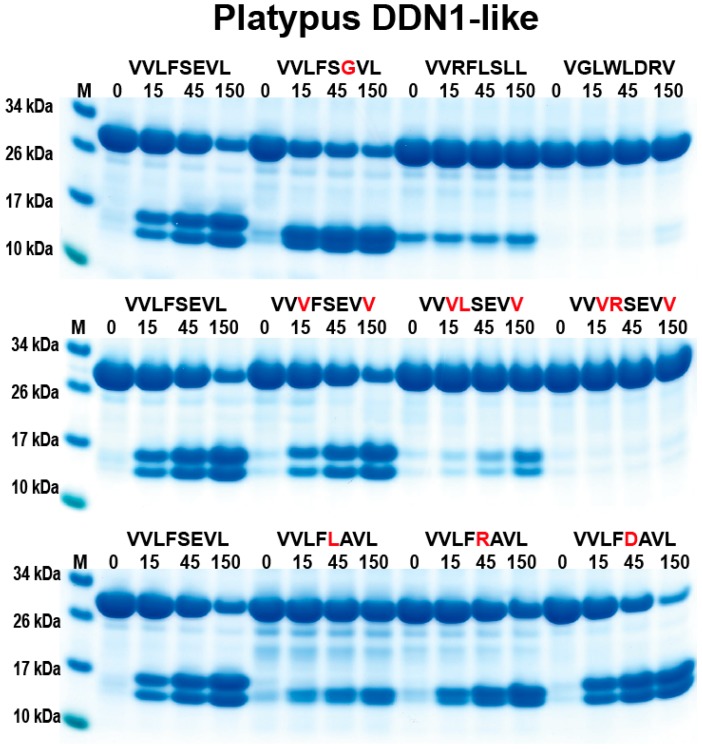
Analysis of the cleavage specificity of platypus DDN1-like (also a chymase) by the use of recombinant protein substrates. The different panels show the cleavage of a number of substrates by the platypus DDN1-like. The sequences of the different substrates are indicated above the pictures of the gels. The red residues highlight the major difference between this substrate and the reference substrate positioned at the beginning (Left side) of this panel. The time of cleavage, in minutes, is also indicated above the corresponding lanes of the different gels. The un-cleaved substrates have a molecular weight of approximately 25 kDa and the cleaved substrates appear as two closely-located bands with a size of 12–13 kDa.

**Figure 6 ijms-21-00319-f006:**
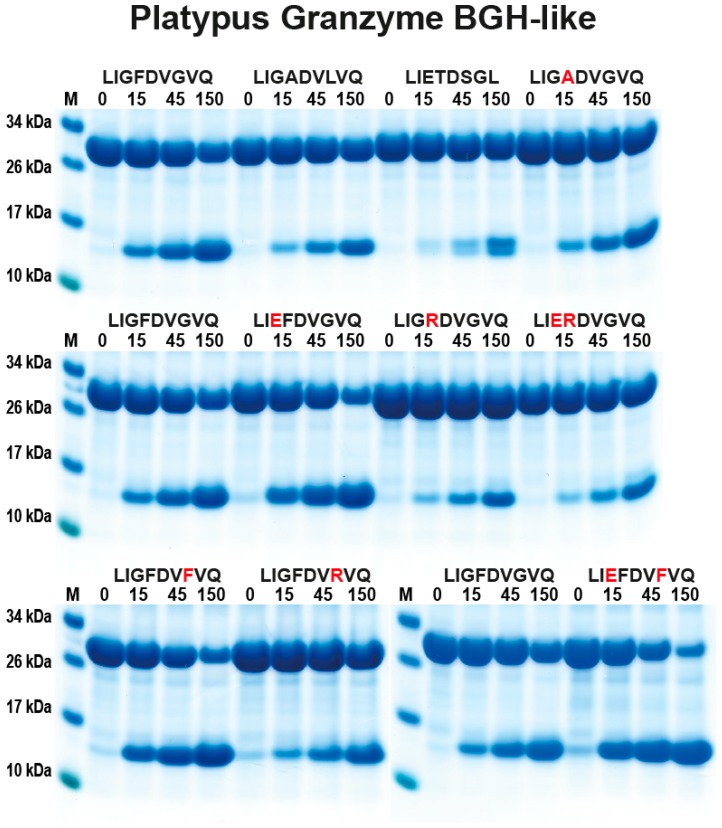
Analysis of the cleavage specificity of platypus GzmBGH (the granzyme B equivalent) by the use of recombinant protein substrates. The different panels show the cleavage of a number of substrates by the platypus GzmBGH. The sequences of the different substrates are indicated above the pictures of the gels. The red residues highlight the major difference between this substrate and the reference substrate positioned at the beginning (Left side) of this panel. The time of cleavage, in minutes, is also indicated above the corresponding lanes of the different gels. The un-cleaved substrates have a molecular weight of approximately 25 kDa and the cleaved substrates appear as two closely-located bands with a size of 12–13 kDa.

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
