# Peer review of "Extended Cleavage Specificities of Two Mast Cell Chymase-Related Proteases and One Granzyme B-Like Protease from the Platypus, a Monotreme"

_ijms, 2020, doi:10.3390/ijms21010319_

Round 1
Reviewer 1 Report
This is a well-designed study that provides with a thorough analysis of the substrate specificities of the platypus "granzyme B" (the chymase), platypus DDN1-like and the platypus GzmBDH. The authors found that platypus "granzyme B" and platypus DDN1-like are classical chymases with specificities very similar to human and mouse mast cell chymases, while platypus GzmBDH has a similar specificity to granzyme B of placental mammals.
Minor comments:
1-Fig. 3: can you please point at the bands that correspond to the synthesized proteases +/- EK?
2-Can the authors summarize their substrate specificity findings using a table in addition to what is described in the text?
3-Did the authors attempt to inhibit the chymotryptic or granzyme activity of the platypus proteases?
Author Response
Reviewer 1.
The bands representing the protease before and after enterokinase cleavage have been marked by small red arrow heads. A sentence clarifying the presence of the red arrow heads has also been added to the figure legend (Marked in red). We would very much have loved to add a table. However, the number of substrates included in this analysis is to small to give a complete picture of the specificity of these enzymes, and as mentioned we have not succeeded in performing a phage display analysis of these three enzymes by unknown reasons despite several attempts. A phage display or a very large number of substrates is in general needed to be able to obtain a picture with sufficient resolution for specifying each residue as we see with phage display where we can produce an Ice-logo type of preference. We would need to analyze each residue with a panel of 4-5 different amino acids, which would involve at least 30-40 additional substrates for each protease and we have not yet such info why a table would be very incomplete and possibly misleading-Sorry. No attempts to inhibit the enzyme has been done which would be interesting to study the control of their activity in vivo. However, no platypus protease inhibitors have been cloned and produced as recombinant proteins to my knowledge and the enzymes are very pure from other contaminating enzymes as they are purified on Ni chelating IMAC columns from a medium containing large amounts of protease inhibitors. Without entrerokinase cleavage we see no enzyme activity of these purified recombinant enzymes.
Reviewer 2 Report
I would like to refer to the article by Powers JC et al published in Biochemisty 1985;24(8): 2048-58. They demonstrated that an efficient substrate for mammalian chymases has the sequence of -Val-Pro-Phe-. Do the authors have experience with proline in the P2 position ?
Fig. 2: I am not sure whether the color codes are correct when looking at the figure and then reading the figure legend
Is it granzyme BDH or granzyme BGH, because both versions are written in the manuscript ?
Some information on the purity level of enzymes would be recommended, because in Fig 3 there are several bands
Fig. 4: There is text on "Kinker region", should it be "-Linker region". Legend text 4 (line 155), should it be 2nd thx, not first one ?
Author Response
Reviewer 2.
By a large number of studies with phage display of a panel of mammalian chymases we have been able to get a quite good picture of the selectivity of these enzymes and in these studies we have not observed a preference for Pro in the P2 position of human, mouse, rat, hamster or opossum chymases indicating that this Pro preference is a phenomenon originating from the use of short chromogenic substrates which lack C-terminal sequences of the cleavage site. Look at Hellman and Thorpe Biol. Chem 2014.
The color code has been corrected. Thanks a lot for observing this error.
Now BGH in all positions. Also here thanks a lot for observing this error.
The few contaminating bands originate from the fetal calf serum of the conditioned media from which we purify the recombinant protein and the major contaminating band is BSA. We have added a sentence concerning this info to the figure legend (Marked in red).
It is correct with kinker region as this is a short flexible Ser-Gly repeating sequence within the linker region. A sentence explaining this has been added to the materials and methods section (Marked in red). Correct the His tag is positioned in the second Trx. This has been modified. Thanks a lot also for finding this error.